# From Apple Waste to Antimicrobial Solutions: A Review of Phenolics from PGI ‘Maçã de Alcobaça’ and Related Cultivars

**DOI:** 10.3390/molecules30183679

**Published:** 2025-09-10

**Authors:** Jessica Ribeiro, Vanessa Silva, Maria de Lurdes N. E. Dapkevicius, Gilberto Igrejas, Lillian Barros, Sandrina A. Heleno, Filipa S. Reis, Patrícia Poeta

**Affiliations:** 1Microbiology and Antibiotic Resistance Team (MicroART), Department of Veterinary Sciences, University of Trás-os-Montes and Alto Douro (UTAD), 5000-801 Vila Real, Portugal; jessicalribeiro97@gmail.com (J.R.); vanessasilva@utad.pt (V.S.); 2Associated Laboratory for Green Chemistry (LAQV), Department of Chemistry, Faculty of Science and Technology, University NOVA of Lisbon, 2829-516 Caparica, Portugal; 3Centro de Investigação de Montanha (CIMO), La SusTEC, Instituto Politécnico de Bragança (IPB), 5300-253 Bragança, Portugal; lillian@ipb.pt (L.B.); sheleno@ipb.pt (S.A.H.); freis@ipb.pt (F.S.R.); 4Department of Genetics and Biotechnology, University of Trás-os-Montes and Alto Douro (UTAD), 5000-801 Vila Real, Portugal; gigrejas@utad.pt; 5Functional Genomics and Proteomics Unit, Department of Genetics and Biotechnology, University of Trás-os-Montes and Alto Douro (UTAD), 5000-801 Vila Real, Portugal; 6Faculty of Agricultural and Environmental Sciences, University of the Azores, 9700-042 Angra do Heroísmo, Portugal; maria.ln.dapkevicius@uac.pt; 7Institute of Agricultural and Environmental Research and Technology (IITAA), University of the Azores, 9700-042 Angra do Heroísmo, Portugal; 8Associate Laboratory for Animal and Veterinary Science (AL4AnimalS), University of Trás-os-Montes and Alto Douro (UTAD), 5000-801 Vila Real, Portugal; 9Veterinary and Animal Research Centre (CECAV), University of Trás-os-Montes and Alto Douro (UTAD), 5000-801 Vila Real, Portugal

**Keywords:** antimicrobial resistance, antibacterial activity, antioxidant activity, *Malus domestica*, bioactive phytochemicals, fruit by-products, oxidative stress, polyphenols

## Abstract

Apple by-products represent a valuable source of phenolic compounds with significant antimicrobial potential, aligning with sustainable strategies for waste valorisation within the circular bioeconomy. This review focuses on the phenolic profile and antimicrobial relevance of ‘Maçã de Alcobaça,’ a Protected Geographical Indication (PGI) apple variety from Portugal. The main phenolics identified include phloridzin, phloretin, chlorogenic acid, quercetin glycosides, catechin, epicatechin, and procyanidins, which exhibit broad-spectrum antibacterial activity, particularly against Gram-positive pathogens such as *Staphylococcus aureus*. Their structure–activity relationships and mechanisms of action, namely membrane disruption, enzyme inhibition, oxidative stress induction, and quorum sensing interference, are discussed. Different extraction methods and solvents influence phenolic yield and bioactivity, with ethyl acetate and hydromethanolic extracts generally showing stronger effects. Studies reveal the potential of phenolics to interact synergistically with antibiotics and the promising applications in food preservation, medical formulations, and antimicrobial packaging. Overall, apple-derived phenolics, particularly those derived from industrial by-products, have significant potential as natural antimicrobial agents. Further exploration of these phenolics in the context of One Health and antimicrobial resistance mitigation is recommended.

## 1. Introduction

The continuous growth of the agri-food industry has led to a significant increase in fruit production. However, this expansion has also resulted in the generation of large amounts of waste, raising environmental and economic concerns. Between 2010 and 2023, global fruit production increased by 30%, reaching a total of 952 million tons, with apples (*Malus domestica*) ranking third in worldwide fruit production (97 million tons), following bananas (139 million tons) and watermelons (105 million tons) [1]. Apples are not only widely consumed fresh but are also processed into various products such as juice, cider, and vinegar, which generate substantial by-products, particularly apple pomace [2,3].

‘Maçã de Alcobaça’ is a Portuguese Protected Geographical Indication (PGI) product that, currently, includes six apple varieties: Fuji, Golden Delicious, Granny Smith, Jonagold, Reineta and Royal Gala [4]. This designation enhances the economic value of ‘Maçã de Alcobaça’ products and contributes to regional economic sustainability by promoting local agricultural practices and tourism (Figure 1) [5]. Despite their high nutritional value and distinctive characteristics, these apples, like other fruit crops, generate a large volume of underutilized residues, primarily from processing industries [6].

Fruit-based industries face growing challenges related to sustainability, as a significant portion of fruit biomass is discarded as waste [7]. In apple processing, pomace, which is composed of peels, seeds, and residual flesh, represents about 25–30% of the original fruit weight [2,6,8]. Although often considered waste, these by-products contain valuable bioactive compounds, particularly phenolic compounds, which are secondary metabolites with diverse biological activities. Their presence in apple residues emphasizes the potential for waste valorization in a circular economy context [6,7].

Phenolic compounds play a crucial role in plant defense mechanisms, acting as antioxidants, antimicrobials, and signaling molecules involved in stress responses [9]. Chemically, they can be classified into several groups, including flavonoids, phenolic acids, tannins, lignans, and stilbenes, with variations in their chemical structures influencing their bioactivity [10]. Among them, flavonoids and phenolic acids are the most abundant in apples, contributing to their antioxidant capacity and potential health benefits [3,11]. In addition to their well-documented roles in preventing diseases related to oxidative stress and modulating the gut microbiota, recent studies have suggested that they have promising antimicrobial properties against drug-resistant pathogens [12,13].

The circular bioeconomy framework considers food waste as a valuable source of high-value phytochemicals, offering a sustainable alternative to synthetic additives [14]. In this context, the valorization of ‘Maçã de Alcobaça’ residues through the extraction of phenolic compounds could provide a dual benefit: reducing environmental impact and offering natural bioactive compounds with potential health applications.

Despite growing interest in apple-derived phenolics, the phenolic composition of ‘Maçã de Alcobaça’ apples and their antimicrobial potential remain under characterized, limiting their broader application in sustainable solutions. Only a few investigations have examined the phytochemical characteristics of these apples, with most data available concerning general *Malus domestica* cultivars. Therefore, this review integrates existing knowledge from both PGI and non-PGI apple studies to highlight the potential of Alcobaça apples in antimicrobial and antioxidant applications, while underscoring the urgent need for targeted research on these Portuguese-grown cultivars.

## 2. Methodology Research

A comprehensive literature review was conducted using ScienceDirect (https://www.sciencedirect.com/) was accessed from 1 February to 30 April 2025 to identify all relevant studies. The search terms included ‘Maçã de Alcobaça’, ‘*Malus domestica*’, ‘phenolic compounds’, ‘antioxidant activity’, and ‘antibacterial activity’. The review is composed of articles that have undergone the peer-review process, with supplementary references drawn from books and reports, when pertinent. The present study excluded conference abstracts and publications in languages other than English. The identification of additional references was accomplished by meticulously examining the reference lists of articles.

## 3. Phenolic Profile of ‘Maçã de Alcobaça’ and Related Apple Cultivars

Standardizing the extraction of phenolic compounds is a complex task due to the difficulty in controlling solvent interactions and operating parameters. To obtain unaltered fractions of the target compounds and maximize yield, it is necessary to achieve an optimal balance between solubilization and degradation [14]. The phytochemical composition of PGI ‘Maçã de Alcobaça’ remains largely uncharacterized, making it essential to investigate its phenolic profile and gain insights into terroir-specific nutrient composition. This review outlines the technical strategies used to overcome the complexities involved in extracting phenolic compounds from these apples.

To date, only two studies have addressed this gap (Table 1). Almeida et al. focused on individual flavonoids in the peel of six PGI ‘Maçã de Alcobaça’ cultivars (Fuji, Golden Delicious, Granny Smith, Jonagold, Reineta, and Royal Gala), reporting significantly higher phenolic yields (14.7–65.1 g GAE kg^−1^) and identifying flavonols such as epicatechin, hyperin, procyanidins, and quercetin derivatives [15]. In this study, phenolics were extracted from 1 g of sample using methanol acidified with 1% HCl, followed by Folin–Ciocalteu quantification. In contrast, Teixeira et al. analyzed the phenolic content in the peel, flesh, and seeds of five traditional Portuguese cultivars from the Alcobaça region (‘Pêro de Borbela’, ‘Pardo Lindo’, ‘Repinau’, ‘Pêro Coimbra’, and ‘Noiva’), identifying chlorogenic acid, vanillic acid, and rutin among the key compounds, with relatively low yields in the peel (0.7–2.0 g GAE kg^−1^) [7]. Here, phenolics were extracted by adding 2 g of powdered sample to 10 mL of MeOH:H_2_O:formic acid (49.95:49.95:0.10, *v*/*v*/*v*) followed by sonication at room temperature for 10 min prior to UHPLC analysis. These variations may be attributed to differences in cultivar selection, extraction methods, or environmental factors affecting phenolic accumulation. The efficiency and yield of phenolics may be further influenced by matrix composition, specifically the relative abundance of fibers and lipophilic compounds [16]. Despite the limited research, these findings highlight that Alcobaça apples are a source of bioactive compounds, particularly the peel, which exhibits the highest phenolic diversity and concentration. Further studies are needed to explore the impact of terroir, maturation stage, and post-harvest conditions on the phenolic composition and potential health benefits of these apples.

The phenolic profile of apples is significantly influenced by cultivar type, tissue compartment, and fruit ripeness [17,18]. Although both the peel and flesh contain phenolic compounds, as previously observed, the peel generally exhibits higher total concentrations [9,19]. Despite some variability among cultivars, chlorogenic acid, catechin, epicatechin, procyanidins, quercetin glycosides, rutin, and phloridzin are consistently identified in apple tissues [2,9,18]. Table 2 categorizes the main phenolic compounds identified in apple extracts according to their phenolic group and provides their chemical structures.

Phloridzin is the predominant polyphenol found in apple seeds and stems (also known as apple peduncles), while the flesh is primarily rich in chlorogenic acid and flavonol glycosides (Figure 2) [21]. In seeds, the most abundant phenolic compounds include chlorogenic acid, (−)-epicatechin, and phloridzin [22]. In the apple flesh, chlorogenic acid stands out as the main hydroxycinnamic acid, while flavan-3-ols such as (+)-catechin, (−)-epicatechin, and procyanidin B2 are also present in considerable amounts. Dihydrochalcones like phloridzin are additionally detected in the flesh, though in lower concentrations compared to the seeds [23,24]. For example, in ‘Golden Delicious’ and ‘Red Delicious’ apples, catechin and epicatechin together account for approximately 32% and 9% of the total phenolic content, respectively; procyanidin B1 contributes around 14% and 26%; and phloridzin/phloretin represent 15% and 12% of the total phenolics. In both cultivars, chlorogenic acid is consistently the most abundant phenolic compound, comprising nearly 40% of the total phenolic content at harvest [18]. The apple peel, in contrast, is especially rich in quercetin glycosides, rutin, and phloridzin [25]. The apple stem, although less studied, shares a similar profile with the seeds, containing high levels of phloridzin and procyanidins [21].

Some cultivars, such as Fuji, Golden Delicious, Granny Smith, and Pink Lady, have higher concentrations of phenolic acids in their flesh than in their peel [26]. This pattern may be related to specific distribution of phenolic compounds in different cultivars and differences in tissue composition, as certain phenolic acids may accumulate in the flesh to provide antioxidant protection or support fruit development [27]. Chlorogenic acid is the only phenolic compound that is more abundant in the flesh than in the peel. Cultivars such as Reinette, Casa Nova, and Starking have flesh concentrations of chlorogenic acid that exceed 1 g kg^−1^, whereas Granny Smith has significantly lower levels (~0.1 g kg^−1^) [15].

Peel pigmentation also appears to influence phenolic composition. Red- to dark-red-skinned apples tend to contain higher levels of chlorogenic and caffeic acids in the peel, while peels of green-skinned cultivars generally exhibit higher concentrations of procyanidins and flavanols [28]. Furthermore, the phenolic profile of fresh apple pomace, a significant byproduct of juice production, is dominated by chlorogenic acid, caffeic acid, (+)-catechin, (−)-epicatechin, rutin, and quercetin glycosides, highlighting the potential of apple residues as a rich source of bioactive compounds [29].

## 4. Biological Activities of Phenolics from PGI ‘Maçã de Alcobaça’ and Related Cultivars

### 4.1. Antibacterial Activity

Phenolic compounds are synthesized via the phenylpropanoid pathway as part of the plant’s defense system and play a crucial role in resistance to microbial infections. In apples, this pathway plays a significant role in disease resistance [30]. Due to their membrane-disruptive properties and ability to interfere with microbial enzymatic systems, phenolic compounds are being studied more extensively as potential solutions for fighting bacterial resistance [6]. Their efficacy is influenced by several factors, including structural differences between Gram-positive and Gram-negative bacteria, the concentration and chemical nature of the phenolic compound, the extraction method employed, and the duration of bacterial exposure [10]. Natural antibacterial agents have garnered increasing attention over the past decade due to the growing need for alternatives to synthetic preservatives in food and pharmaceutical products [6,31]. Among these, phenolic compounds are particularly promising due to their broad-spectrum antimicrobial activity and relatively low toxicity [6,32].

#### 4.1.1. Mechanisms of Bacterial Inhibition

To better understand the antibacterial potential of apple-derived phenolics, it is essential to explore the underlying mechanisms by which these compounds exert their effects. Phenolic compounds exert their antimicrobial activity through diverse and often interconnected mechanisms (Figure 3).

One of the primary actions is the disruption of the bacterial cell membrane. By increasing membrane permeability, many phenolics cause leakage of ions, ATP, and other essential intracellular components, compromising cell viability [33]. Additionally, some phenolic compounds are known to interfere with intracellular targets, particularly enzymes involved in vital processes such as nucleic acid synthesis, energy metabolism, and oxidative stress regulation [10]. Another relevant mechanism is the induction of oxidative damage. Under specific conditions, phenolics can act as pro-oxidants by promoting the generation of reactive oxygen species (ROS), leading to oxidative stress and damage cellular structures including proteins, lipids, and DNA. This contributes significantly to their bactericidal effect [34]. Also, certain phenolics can inhibit quorum sensing and biofilm formation, thereby preventing bacterial communication and collective behaviors essential for pathogenicity and antibiotic resistance. By interfering with signaling molecules, these compounds can reduce virulence and enhance susceptibility to conventional treatments [10,34]. Recent evidence also indicates that phenolic compounds may modulate bacterial efflux pumps and resistance gene expression. Studies have shown that phenolics, such as quercetin, and epigallocatechin gallate, have downregulated efflux pump genes or other resistance determinants, thereby restoring or enhancing the efficacy of antibiotics [32].

A fascinating aspect of phenolic compounds is their ability to inhibit various types of pathogenic bacteria, such as *E. coli*, *Pseudomonas aeruginosa*, *S. aureus*, and *L. monocytogenes*, while simultaneously promote the growth of beneficial microbes such as *Lactiplantibacillus plantarum*, *Saccharomyces boulardii*, and *Bifidobacterium* spp. [35]. The bifidogenic effect has previously been reported, suggesting that phenolic compounds, including catechin and epicatechin, may stimulate the growth of beneficial gut bacteria, thereby contributing to the modulation of the gut microbiota and promoting host health [36,37,38]. This dual effect appears to depend on both concentration and structure, possibly reflecting differences in how microbes handle phenolic compounds or their ability to handle oxidative stress [10].

#### 4.1.2. Structure-Activity Relationship

The antimicrobial activity of phenolic compounds is closely related to their chemical structure, including the number and position of hydroxyl groups, the presence of glycosidic moieties, and the overall hydrophobicity or lipophilicity (Table 3) [39,40].

Among the phenolic constituents found in apples, phloridzin and phloretin are the most studied in terms of antibacterial activity. Phloridzin is a glucoside of phloretin, with a glucose moiety attached at the 2′ position of the B-ring. This glycosylation increases polarity and stability but reduces antimicrobial potency [41]. By contrast, phloretin lacks the glucose unit, exhibits greater lipophilicity, and penetrates bacterial membranes more effectively, particularly Gram-positive bacteria [45]. This structural distinction is reflected in their MICs, with phloretin frequently demonstrating stronger antimicrobial activity against Gram-positive and Gram-negative bacteria [46]. Phloridzin and phloretin also exhibit differing mechanisms of action. Phloretin has been shown to disrupt bacterial membranes, alter membrane fluidity, and inhibit biofilm formation in *Escherichia coli* O157:H7 while sparing commensal strains such as *E. coli* K-12 [42]. Phloridzin, although less potent, can act as a membrane penetration enhancer, potentially increasing uptake of co-administered antimicrobial agents [46].

Other phenolic compounds, including chlorogenic acid, catechins, and procyanidins, contribute to antibacterial activity through varied mechanisms. Chlorogenic acid primarily acts by disrupting bacterial membranes, and generating ROS, which can damage DNA, proteins, and lipids, showing stronger effects against Gram-positive bacteria while retaining some activity against Gram-negative strains [47,48]. Catechins, such as epicatechin and epigallocatechin derivatives, exert antibacterial effects through binding to membrane proteins, enzyme inhibition, and alterations of membrane permeability, and are also able to inhibit bacterial adhesion and biofilm formation [49]. The activity of catechins is influenced by structural features, including hydroxylation patterns, with epigallocatechin and epigallocatechin gallate often showing enhanced activity against *E. coli* and *S. aureus* [50]. Procyanidins, abundant in apple skin extracts, exhibit notable antibacterial activity, which correlates with their degree of polymerization and hydrophilic side chains [32,43].

Quercetin is often present as glycosides such as quercetin-3-glucoside and rutin. Glycosylation increases the polarity of the molecule and reduces membrane permeability compared with the aglycone, generally resulting in lower direct antibacterial potency [51]. The planar structure of quercetin enables it to interact with bacterial enzymes and membrane components. These interactions contribute to the inhibition of key bacterial enzymes and the generation of ROS, which can damage bacterial DNA, proteins, and membranes [49]. Accordingly, quercetin aglycone was found to be more active than rutin against methicillin-resistant *S. aureus*. Additionally, it was found that when used in combination with antibiotics, they increased each other’s activity [52].

In general, increased hydrophilicity from hydroxyl or glycosidic substituents can enhance aqueous solubility but may reduce membrane permeability, whereas more lipophilic structures tend to exhibit greater bactericidal activity due to improved membrane affinity [51]. Understanding these structure–activity relationships is essential to guide the selection and optimization of apple-derived phenolics as natural antimicrobials in food preservation, nutraceuticals, and pharmaceutical formulations.

#### 4.1.3. Extraction-Activity Relationship

The antimicrobial activity of phenolic-rich extracts is also influenced by the extraction method, which determines the yield, composition, and stability of the phenolic compounds obtained [53]. Solvent polarity, extraction temperature, time, and the use of novel techniques (e.g., ultrasound-assisted extraction, microwave-assisted extraction, pressurized liquid extraction) can significantly impact the bioactivity of the resulting extracts. Particularly, polar solvents such as methanol and ethanol often yield higher concentrations of phenolic acids and flavonoids, which are key to antibacterial activity [54]. Additionally, novel extraction methods have been shown to increase the yield of bioactive compounds compared to conventional maceration or Soxhlet extraction. However, thermal degradation, solvent toxicity, and matrix interactions must be considered when selecting an extraction protocol [55]. It is also important to recognize the synergistic or antagonistic effects that may arise from co-extracting multiple phenolic compounds. The antimicrobial profile of the extract is often the result of the combined and individual activities of the compounds (see Table 4) [10].

Zhang et al. have evaluated the antibacterial activities of the ethyl acetate extract, phloridzin, and phloretin. The ethyl acetate extract exhibited notable inhibitory effects against *Staphylococcus aureus* (MIC = 1.25 mg/mL) and *E. coli* (MIC = 2.50 mg/mL). Phloridzin and phloretin showed superior inhibitory effects, with MICs of 0.50 and 0.10 mg/mL, respectively, against *S. aureus*, and 1.50 and 0.75 mg/mL, respectively, against *E. coli* [46]. Phloridzin has been described as a drug penetration enhancer due to its ability to bind biological membranes and increase their fluidity [9]. Both phloridzin and its derivatives exhibit significant antimicrobial activities [41,56]. According to Lee et al., phloretin inhibited *E. coli* O157:H7 biofilm formation while preserving beneficial commensal *E. coli* K-12 biofilms [42].

In a study focused on the antimicrobial potential of apple leaf extracts, dried apple leaves were subjected to triple maceration with 70% aqueous ethanol. The pooled extract was subsequently fractionated through liquid–liquid partitioning using hexane and ethyl acetate, yielding three fractions: hexane, ethyl acetate, and aqueous. Among these, only the hexane fraction exhibited relevant antibacterial activity, with a MIC of 1.18 mg/mL against *B. subtilis*, *K. pneumoniae*, *S. aureus*, *M. luteus*, and *E. coli*, and a MIC of 2.37 mg/mL against *L. monocytogenes* [56]. These findings suggest that non-polar compounds in apple leaves may contribute to the observed antimicrobial effects.

Hydromethanolic extracts from the dried Portuguese apple variety ‘Bravo de Esmolfe’ showed the lowest MIC values against Gram-positive bacteria, namely methicillin-susceptible *S. aureus* (MSSA) (MIC = 2.5 mg/mL). MRSA and other Gram-positive bacteria, such as *Listeria monocytogenes* and *E. faecalis*, displayed higher MIC values (MIC = 5 mg/mL). Among Gram-negative bacteria, ESBL *-E. coli*, and *M. morganii* were the most susceptible, with MICs also of 5 mg/mL [13].

A combined approach involving organic solvent extraction and solid-phase extraction was applied to sliced apple matrix using a 25:75 (*v*/*v*) acetone:ethanol solution. This method produced an unfractionated phenolic extract and four phenolic fractions that were then evaluated for antimicrobial activity. The fractions exhibited notable inhibitory effects against *S. aureus*, *E. coli*, *Listeria monocytogenes*, and *S. typhimurium*, with MIC values ranging from 0.05 to 0.5 mg/mL for most strains. *L. monocytogenes* showed higher variability in susceptibility across the four phenolic fractions. The highest MIC was observed for fraction IV, reflecting differences in its phenolic composition compared to the other fractions [57].

Moreover, the antibacterial activity of food-based powders against food-borne pathogens was investigated. A 4% stock solution of apple skin extract was prepared and serially diluted to assess its bactericidal activity against *E. coli* O157:H7, *L. monocytogenes*, *Salmonella enterica*, and *S. aureus*. The concentrations required to kill 50% of the pathogens were >2.7%, 1.39%, 0.007%, and 0.002%, respectively. The apple skin extract exhibited particularly strong growth inhibition against *S. aureus*. Procyanidins and phloridzin were identified as major bioactive constituents in the extract [43].

In summary, phenolic compounds from apples, particularly phloridzin, and phloretin, have shown notable antibacterial activity, especially against *S. aureus* and other Gram-positive pathogens. For comparison, conventional antibiotics display MIC values in the low µg/mL range: ciprofloxacin (5 µg) between 0.004–0.016 µg/mL for *E. coli* and 0.125–0.5 for *S. aureus*; vancomycin (1 µg) between 0.5–2 µg/mL for *S. aureus*; gentamicin (10 µg) between 0.25–1 µg/mL for *E. coli* [58]. Apple phenolics such as phloretin and phloridzin demonstrated higher MIC values than conventional antibiotics, indicating lower potency, although still significant for natural compounds. This highlights their potential as adjuvants or food preservatives instead of individual therapeutic agents. The efficacy depends on the extraction method, compound polarity, and bacterial strain. While in vitro results are promising, further studies are needed to clarify mechanisms of action, assess synergistic potential with antibiotics, and confirm efficacy in vivo.

### 4.2. Antioxidant Activity

Almeida et al. evaluated the antioxidant activity of both peel and flesh extracts from several Alcobaça apple cultivars (‘Casa Nova’, ‘Gala’, ‘Granny Smith’, ‘Reinette’, ‘Starking’, ‘Golden’, ‘Fuji’, and ‘Jonagored’) using the ABTS radical scavenging assay and complementary in vitro biological methods, including inhibition of DNA oxidative damage and protection against H_2_O_2_-induced oxidation of phage P22. Their results consistently showed that peel extracts had higher antioxidant activity: the peel extracts had almost five times more antioxidant activity than the flesh, reinforcing the recommendation to consume apples unpeeled. Among the tested cultivars, ‘Casa Nova’ and ‘Reinette’ displayed the highest antioxidant activity in the flesh, while ‘Starking’ stood out for the peel. In contrast, ‘Golden Delicious’ and ‘Granny Smith’ showed the lowest antioxidant activity in peel extracts [44].

In a follow-up study, Almeida et al. investigated methanolic and acetonic phenolic extracts of eight Alcobaça apple cultivars (‘Casa Nova’, ‘Fuji’, ‘Golden Delicious’, ‘Granny Smith’, ‘Jonagored’, ‘Reinette Grise’, ‘Royal Gala’, and ‘Starking’) using the ABTS assay. On average, the antioxidant activity of the eight cultivars was higher in the methanolic extract than in the acetone extract. The flesh and peel had 10- and 5-fold higher activity, respectively [15]. The apple peels always showed higher antioxidant activity phenolics (1.78–5.49 mg g^−1^), than the flesh phenolics (0.34–1.06 mg g^−1^), which agrees with other studies [15,59]. Consistently, across all cultivars, the peel contained larger amounts of total phenolics (14.7–65.1 g GAE kg^−1^ FW), compared to the flesh (5.2–14.4 g GAE kg^−1^ FW). No direct correlation was found between apple color and antioxidant capacity [15].

Teixeira et al. expanded this line of investigation by analyzing five Alcobaça-region cultivars using DPPH and β-carotene bleaching assays, along with assessments of total phenolic and flavonoid contents. Their two-year study (2021–2022) pointed to the impact of edaphoclimatic conditions and agricultural practices on antioxidant capacity. Among their findings, the ‘Noiva’ cultivar exhibited the highest antioxidant activity in the peel, with values ranging from 648 to 670 μg Trolox Equivalents (TE)/g FW). In contrast, the seeds of ‘Pêro de Borbela’ cultivar harvested in 2022 showed the strongest antioxidant potential, reaching 211 μg TE/g FW. Accordingly, the highest phenolic content was also detected in the peel of ‘Noiva’ (1964 μg GAE/g FW), whereas in seeds, ‘Pêro de Borbela’ accumulated the greatest amounts (782.6 μg/g FW). In this case, phlorizin was the predominant compound (318.7 μg/g FW), followed by chlorogenic acid (293.5 μg/g FW) and epicatechin (59.31 μg/g FW). Overall, by-products such as peel and seeds consistently exhibited much higher antioxidant activity and phenolic abundance than the pulp [7].

In a different approach, Bottu et al. explored the antioxidant potential of apple pomace extracts obtained through deep eutectic solvent (DES) extraction. The authors used a DES composed of choline chloride and ethylene glycol in a 1:4 molar ratio, which proved highly effective for extracting bioactive compounds. Using DPPH and FRAP assays and evaluating insulin secretion from pancreatic β-cells, they found that DES-extracted pomace showed stronger antioxidant properties, which highlights the relevance of DES systems for efficiently extracting bioactive compounds with potential health applications. Furthermore, the drying of the biomass had an impact on the composition of the extracts and the antioxidant capacity. The antioxidant capacity was higher in extracts from dry apple pomaces using DESs and classical approaches. Notably, the DES extracts from wet biomass did not contain detectable levels of procyanidin B2, chlorogenic acid, epicatechin, vanillin, or phloridzin [60]. These results suggest that the antioxidant potential of pomace extracts is strongly dependent on the presence of key phenolics such as procyanidin B2, chlorogenic acid, epicatechin, vanillin, and phloridzin.

Among relevant compounds, chlorogenic acid, predominant in the peel and flesh of apples, displayed one of the highest antioxidant activities among 18 antioxidant compounds, including quercetin [22,61]. In vivo studies show that this phenolic acid enhanced antioxidant enzyme activities (superoxide dismutase (SOD), glutathione-px (GPx), catalase (CAT) and glutathione reductase (GSH)) in mice under arsenite-induced oxidative stress, highlighting its protective role at the cellular level. Similarly, epicatechin was found to reduce tumor necrosis factor-alpha TNF-α release in an Alzheimer’s disease mouse model, supporting its neuroprotective potential [62].

Overall, these findings demonstrate that the antioxidant potential of Alcobaça apple cultivars is higher in peel, seeds and pomace, aligning with their concentrations of phenolic compounds. The link between higher total phenolic content and some phenolic compounds (e.g., quercetin glycosides, catechin/epicatechin, phloridzin, procyanidins, and chlorogenic acid), strengthens the correlation between chemical composition and antioxidant activity. The use of innovative extraction techniques, such as DES, not only enhances phenolic recovery but also reinforces the potential for sustainable valorization of apple processing residues. Altogether, the phenolic profile of Alcobaça apples, particularly in their peel, supports their promising role as functional food ingredients.

### 4.3. Other Health Benefits

Phenolic compounds found in the Alcobaça apple exhibit a wide spectrum of biological activities that contribute to human health.

Among them, phloridzin has gained considerable scientific attention due to its potent pharmacological properties. This phenolic compound acts as a sodium-glucose co-transporter (SGLT) inhibitor, reducing intestinal glucose absorption and renal glucose reuptake. Through the inhibition of SGLT1 and SGLT2, it shows promising therapeutic potential for prediabetes and type 2 diabetes mellitus (T2DM) [63,64]. It has also been described as an insulin sensitivity enhancer by promoting glucose uptake in peripheral tissues and modulating insulin-signaling pathways, including PI3K/Akt activation [65]. In addition, phloridzin shows strong anti-inflammatory activity which is supported by studies that indicated reduced intestinal inflammation following apple consumption [42,66]. In vitro assays have shown that phloridzin increases intracellular glutathione in Caco-2 and HT-29 cells and suppresses NF-κB and IL-8 expression in DLD1 cells [67,68]. However, bioavailability studies using ileostomy subjects suggest that, although phloridzin reaches the intestinal lumen after apple juice or cider ingestion, its metabolism in the colon remains unclear, which limits current physiological estimates [69,70]. Phloretin, the aglycone form of phloridzin, exerted anti-inflammatory effects in colitis models, supporting the therapeutic versatility of apple phenolics [42]. Nevertheless, despite its broad pharmacological potential, the possible toxicity of phloridzin should also be considered. Preclinical studies have reported dose-dependent gastrointestinal effects, mainly related to its ability to competitively inhibit SGLT1 and SGLT2, which can result in reduced intestinal glucose absorption and osmotic diarrhea at high concentrations [71]. Thus, while phloridzin remains a promising bioactive molecule, careful attention to its safety profile is warranted.

Chlorogenic acid was associated with the restoration of the expression of inflammatory genes, including TNF-α, and interleukins (IL-1b, IL-6, and IL-10), and genes encoding antioxidant enzymes, such as SOD1, SOD2, and GPx1. Furthermore, it has also improved microbiota diversity, hyperglycemia, and hyperlipidemia and reduced hepatic lipid overaccumulation in mice [72].

Key polyphenols such as rutin and quercetin have shown cardioprotective effects both in vitro and in vivo: rutin improved metabolic syndrome parameters in animal models, while quercetin reduces vascular inflammation associated with atherosclerosis in mice in vivo [73,74]. Quercetin was responsible for reducing the expression of human C-reactive protein and cardiovascular risk factors (SAA and fibrinogen) [74]. In addition to these cardiometabolic benefits, quercetin derivates obtained from apple tree leaves exhibited strong neuroprotective properties [75].

Overall, apple peel extracts were found to have superior effects on vascular endothelial function, blood pressure modulation, lipid metabolism, and insulin sensitivity [76]. These compounds’ health-promoting properties highlight their value as functional dietary components, particularly in preventing and controlling cardiometabolic disorders. Taken together, these findings support the diverse potential of Alcobaça apple polyphenols as promising candidates for functional foods and nutraceuticals. Further research should aim to clarify the molecular mechanisms underlying these effects, explore the interactions among polyphenols, and develop optimized extraction systems to enhance bioavailability and therapeutic efficacy.

## 5. Potential Applications of Phenolics from PGI ‘Maçã de Alcobaça’ and Related Cultivars

Phenolic compounds derived from ‘Maçã de Alcobaça’ and other apple cultivars have multifunctional bioactivities, making them promising candidates for various industrial and therapeutic applications. Their natural origin and antimicrobial and antioxidant properties, support their use in medical, food, and pharmaceutical contexts. This section provides an overview of their potential applications, emphasizing their synergistic effects with antibiotics, their utility in food preservation systems, and their prospective roles in therapeutic interventions.

### 5.1. Synergistic Effects with Antibiotics

One of the most compelling applications of apple-derived phenolic compounds is their ability to work together with conventional antibiotics, especially against multidrug-resistant bacteria [77]. Several phenolic compounds such as phloretin, quercetin, and chlorogenic acid have demonstrated the capacity to enhance antibiotic efficacy by targeting bacterial resistance mechanisms. These include the inhibition of efflux pumps, disruption of bacterial membranes, interference with energy metabolism, and suppression of quorum sensing pathways [10,32,41].

An example of this approach is the use of flavonoids to treat β-lactam-resistant *S. aureus*. This approach has yielded encouraging outcomes when used as monotherapy and in conjunction with β-lactam antibiotics. The observed antimicrobial activity was attributable to two factors. First, a change in cell wall integrity was observed, resulting in a significant impairment of protein synthesis. Second, the natural compounds directly inhibited the penicillin-binding protein [34]. Similarly, quercetin, when combined with meropenem, inhibited carbapenems and efflux pumps of carbapenem-resistant Gram-negative bacteria [78]. Chlorogenic acid has demonstrated synergism with tetracycline against six different pathogenic bacteria, resulting in significant reductions in MICs in vitro [79].

Beyond direct antimicrobial enhancement, some phenolics modulate bacterial virulence and downregulate resistance gene expression, offering a multi-targeted strategy against multidrug resistant pathogens [34]. These findings support the exploration of apple phenolics as antibiotic adjuvants, either in oral formulations or topical applications for skin and soft tissue infections. However, further pharmacodynamic and pharmacokinetic studies are needed to assess their safety, optimal dosing, and stability in clinical settings.

### 5.2. Application in Food Systems

The food industry is constantly searching for natural alternatives to synthetic preservatives, and apple-derived phenolic compounds are a sustainable and effective solution [12]. Due to their antioxidant and antimicrobial actions, they can be incorporated into food matrices to extend shelf life, inhibit spoilage microorganisms, and prevent the oxidative degradation of lipids and vitamins [31,61]. For example, the addition of phenolic-rich extracts to fresh cheese has been shown to delay microbial growth without altering sensory characteristics [80]. In meat systems, these compounds have exhibited the ability to inhibit lipid peroxidation and maintain color stability during refrigerated storage [81].

Another promising approach involves the use of phenolic extracts in active packaging. Apple phenolics can be incorporated into biodegradable films or edible coatings made from chitosan, alginate, or starch, creating packaging materials with intrinsic antimicrobial and antioxidant properties. Such systems have proven effective in delaying microbial spoilage of fruits, vegetables, and seafood [82].

Additionally, developing encapsulated phenolic systems, such as nanoemulsions or microcapsules, allows for the controlled release of active compounds and improves solubility in various food matrices. These formulations can be optimized to minimize their impact on taste and appearance while maximizing their bioactivity. However, regulatory approval, scalability, and consumer acceptance remain key factors for their widespread adoption in food systems [83].

### 5.3. Medical and Therapeutic Applications

The therapeutic potential of apple phenolic compounds extends beyond their use as antimicrobials. They also have anti-inflammatory, cardioprotective, and metabolic regulatory properties [22]. There is growing interest in incorporating them into nutraceuticals, medical devices, and dermatological formulations, particularly in alignment with One Health and personalized medicine frameworks [84].

Topically, apple phenolics have shown promise in wound healing and skin barrier protection [85]. Phloridzin and quercetin derivatives exhibit strong anti-inflammatory effects, including pathway inhibition and modulation of cytokine release. These properties make them suitable candidates for incorporation into creams, ointments, and hydrogel dressings [83]. Also, phenolic-coated medical devices are being explored as a means of reducing the risk of device-associated infections by inhibiting microbial adhesion [86].

Oral formulations containing apple phenolics have demonstrated beneficial effects on glycemic control, lipid metabolism, and vascular function [2,61,76]. Clinical and preclinical studies suggest that compounds like phloretin and chlorogenic acid can improve insulin sensitivity, reduce oxidative stress, and modulate gut microbiota [72,87]. Strategies for encapsulating these phenolics using liposomes, cyclodextrins, or polymeric nanoparticles are being developed to enhance their bioavailability and overcome the challenges posed by their poor solubility and rapid metabolism [53].

Additionally, the anti-proliferative and neuroprotective activities of some phenolic compounds provide opportunities for research into cancer prevention and mitigation of neurodegenerative diseases [87]. Formulations tailored to specific therapeutic targets, such as cardiovascular support, diabetes prevention, and cognitive health, may soon integrate apple phenolic compounds as active ingredients, provided adequate in vivo validation and safety profiling are conducted.

## 6. Challenges and Future Directions

Although phenolic compounds derived from PGI ‘Maçã de Alcobaça’ and related cultivars exhibit antioxidant and antimicrobial potential, several challenges must be addressed to allow their broader application in functional foods, pharmaceuticals, and sustainable technologies. One of the main limitations is the lack of scientific studies specifically focused on Alcobaça cultivars. While a few investigations have explored their phytochemical profiles and bioactivities, the available data remains insufficient to support strong comparisons or industrial standardization. This underscores the importance of conducting systematic studies that evaluate phenolic variability among cultivars.

Another critical challenge lies in standardization and improving the efficiency of extraction techniques. Current methods vary considerably in terms of solvents, conditions, and yield, which makes it difficult to compare results or scale up for industrial use. Although innovative, eco-friendly approaches, such as DES systems, have shown promise in improving both recovery and bioactivity, they require further validation and optimization to meet regulatory and commercial requirements.

Moreover, although in vitro assays consistently demonstrate strong antioxidant and antibacterial effects of apple phenolics, their bioavailability and in vivo efficacy remain underexplored. The physiological relevance of compounds such as phloridzin and chlorogenic acid in humans is still not fully understood, as they undergo significant metabolism in the gastrointestinal tract. Clinical trials and advanced delivery systems, such as microencapsulation or nanoformulations, could play an essential role in overcoming these limitations by enhancing stability, absorption, and targeted delivery.

There is also promising yet underexploited potential for synergistic interactions between apple-derived phenolic compounds and conventional antibiotics or other phytochemicals. Understanding these molecular-level interactions could support the development of novel therapeutic strategies to fight antimicrobial resistance. Additionally, integrating phenolic-rich extracts into active food packaging or dermocosmetic formulations creates new opportunities for circular economy models.

Finally, regulatory frameworks, safety assessments, and consumer acceptance must be considered to ensure the successful incorporation of these compounds into the market. Establishing quality standards, toxicological profiles, and functional claims supported by clinical evidence will be crucial to transition from research to application.

Taken together, the phenolic compounds found in ‘Maçã de Alcobaça’ represent a valuable yet underutilized natural resource with high added value. Their multifunctional bioactivities (antioxidant, antimicrobial, anti-inflammatory, and synergistic effects) highlight their potential in promoting human and animal health, enhancing food safety, and contributing to more sustainable and resilient agri-food systems. The valorization of these apples and their by-products exemplify how local biodiversity can be strategically harnessed to address global challenges such as antibiotic resistance, environmental degradation, and the prevention of chronic diseases. Advancing research, innovation, and interdisciplinary collaboration in this field is essential to realizing this potential fully.

## 7. Conclusions

Phenolic compounds extracted from PGI ‘Maçã de Alcobaça’ and related apple cultivars exhibit antimicrobial and antioxidant properties, particularly in underutilized parts such as peels, seeds, pomace, and leaves. These bioactivities are largely attributed to phenolics such as phloridzin, phloretin, chlorogenic acid, quercetin glycosides, catechins, and procyanidins, which act through diverse mechanisms including membrane disruption, oxidative stress induction, enzyme inhibition, and quorum sensing interference. Several extracts have shown the ability to inhibit foodborne and clinical pathogens.

The data presented in this review supports the strategic utilization of apple by-products as a sustainable and functional source of natural antimicrobials, with potential applications in food preservation, active packaging, and therapeutic formulations. However, significant knowledge gaps remain regarding cultivar-specific phenolic profiles, in vivo bio-efficacy, and the optimization of extraction and delivery systems.

Future research should prioritize the standardization of extraction protocols, clarification of structure–activity relationships, and development of stable, bio-accessible formulations. Finally, the valorization of ‘Maçã de Alcobaça’ aligns with circular economy principles and One Health strategies, contributing to innovative approaches in fighting antimicrobial resistance.

## Figures and Tables

**Figure 1 molecules-30-03679-f001:**
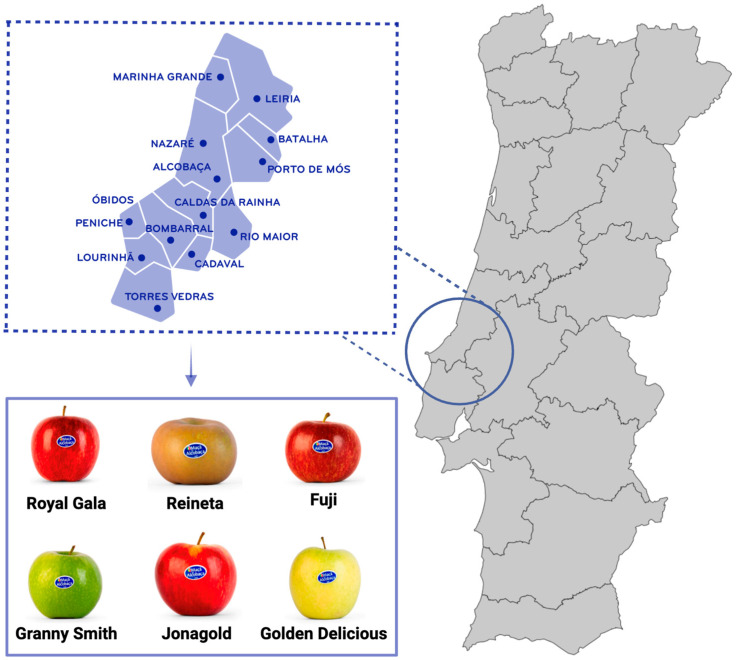
Geographical location of the ‘Maçã de Alcobaça’ production region in Portugal, highlighting the main growing areas of the six varieties included in the Protected Geographical Indication (PGI): Fuji, Golden Delicious, Granny Smith, Jonagold, Reineta and Royal Gala.

**Figure 2 molecules-30-03679-f002:**
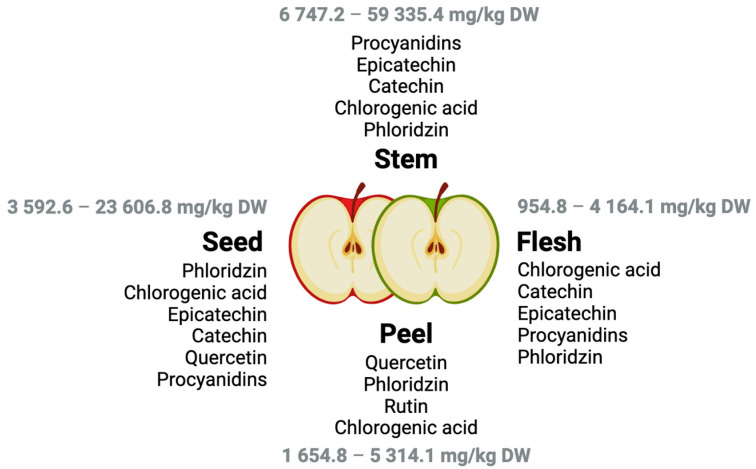
Main phenolic compounds distributed across different anatomical parts of the apple, including seeds, flesh, peel (green and red), and stem [21,22,23,24,25].

**Figure 3 molecules-30-03679-f003:**
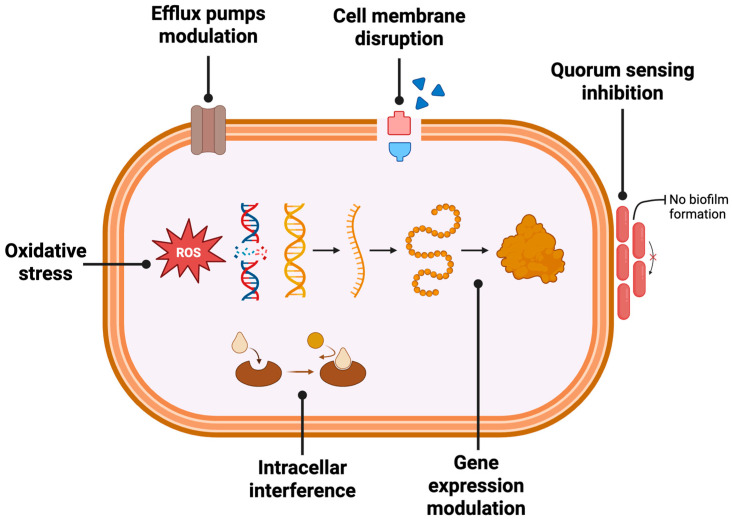
Main antibacterial mechanisms of phenolic compounds: membrane disruption, intracellular interference, oxidative stress, and quorum sensing inhibition.

**Table 1 molecules-30-03679-t001:** Phenolic compounds identified in different matrixes of Alcobaça apples.

Sample (Weight)	Solvent (Volume)	Extract Preparation	Phenolic Compounds	Yield (g GAE kg^−1^ FW)	Ref.
Peel (1 g)	MeOH:H_2_O:HCl (99:0:1 *v/v/v*) (-)	-	Epicatechin Hyperin Procyanidin B2 Phloridzin Quercitrin Catechin Isoquercetin Procyanidin B1 Quercetin Rutin	14.7–65.1	[15]
Flesh (1 g)			-	5.2–14.4	
Peel (2 g)	MeOH: H_2_O: HCOOH (49.95:49.95:0.10 *v*/*v*/*v*) (10 mL)	The solution was agitated for 15 min in a horizontal shaker and centrifuged for 10 min at 20 °C. The supernatant was removed to another Falcon tube, and the extract was repeated with another 10 mL of the solvent. The second extract was merged with the first one.	Chlorogenic acidQuercetin-3-B-d-glucosideEpicatechin Phloridzin 4-Hydroxybenzoic acid Vanillic acid Gallic acid Rutin	0.7–2.0	[7]
Flesh (2 g)			Chlorogenic acid Gallic acid Vanillic acid	0.1–0.4	
Seeds (2 g)			Chlorogenic acid Gallic acid Phloridzin Quercetin-3-B-d-glucoside Vanillic acid Quercetin	0.3–0.9	

FW—fresh weight;-not specified in the original study.

**Table 2 molecules-30-03679-t002:** Chemical structure of phenolic compounds frequently found in apples.

Phenolic Group	Chemical Structure	
Hydroxycinnamic acids Aromatic compounds with the framework of the C6-C3 structure [10].	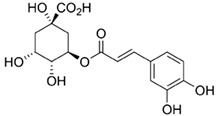 Chlorogenic acid	
Flavanols & Procyanidins A hydroxyl group is attached to position 3 of the ring C, and there is no double bond between positions 2 and 3, nor a carbonyl group at position 4 [20].	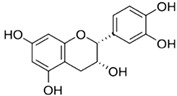 Epicatechin	
Flavonols A hydroxyl group is attached to position 3 of the ring C, and there is a double bond between positions 2 and 3, and a carbonyl group at position 4 [20].	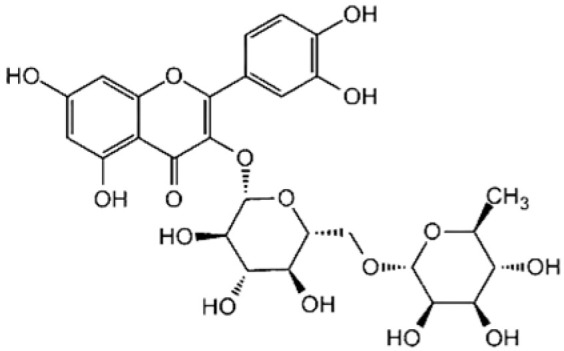 Rutin	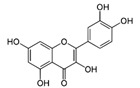 Quercetin
Dihydrochalcones Absence of ring C, consisting of two aromatic rings (ring A and B) linked by an aliphatic three-carbon chain [20].	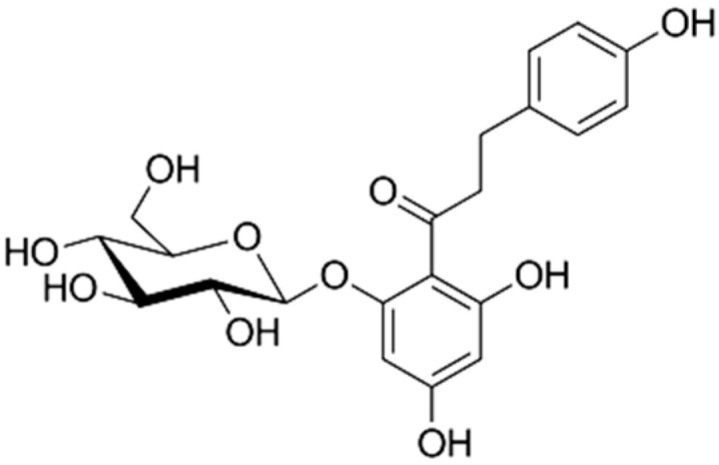 Phloridzin	

**Table 3 molecules-30-03679-t003:** Structure–activity relationship of the main phenolic classes identified in ‘Maçã de Alcobaça’ and related cultivars.

Compound	Phenolic Class	Key Structural Features	Main Antibacterial Mechanism(s)	Observations	Ref.
Phloridzin	Dihydrochalcone	Glucoside of phloretin (sugar at 2′-OH)	Enhances membrane permeability; promotes drug uptake	Less potent than aglycone; acts as penetration enhancer	[41]
Phloretin	Dihydrochalcone	Aglycone (no sugar moiety); lipophilic	Disrupts bacterial membrane; inhibits biofilm; increases fluidity	Stronger activity vs. Gram-positives and *E. coli* O157:H7	[41,42]
Chlorogenic acid	Hydroxycinnamic acid	Caffeic acid esterified with quinic acid	Disrupts membrane; induces oxidative stress	Major phenolic in flesh and pomace	[2,10]
Catechin/Epicatechin	Flavanols	Multiple hydroxyls; flexible structure	Binds to membranes and proteins; causes oxidative stress	Present in both peel and flesh; activity varies by configuration	[11,34]
Procyanidins	Flavanol oligomers	Catechin dimers (e.g., B2); degree of polymerization critical	Membrane destabilization; enzyme inhibition	Abundant in skin; higher polymerization may enhance activity	[13,32,43]
Quercetin glycosides (e.g., quercetin-3-glucoside, rutin)	Flavonols	Glycosylated quercetin; planar structure	Enzyme inhibition; ROS generation; lower permeability than aglycone	Activity affected by glycosylation; quercetin more active than rutin	[15,44]

**Table 4 molecules-30-03679-t004:** Some extraction methods used for phenolic compounds from apples with corresponding antibacterial activities.

Method	Sample (Weight)	Solvent (Volume)	Extract Preparation	Yield	Tested Microorganisms	MIC (mg/mL)	Ref.
Ultrasound-assisted extraction	Pomace (100 g)	100% ethyl acetate (500 mL)	The solution was placed in an ultrasonic bath at 37 °C for 40 min.	2.51 g GAE kg^−1^ DW	*S. aureus*	1.25	[46]
					*E. coli*	2.50	
Maceration + liquid–liquid partition	Leaves (100 mg)	70:30 ethanol:water (*v*/*v*) + hexane (2 mL)	The solution was vortexed for 2 min followed by centrifugation for 10 min at room temperature. The extraction process was repeated twice with 1.5 mL solvent and supernatant was collected and pooled to make final volume 5 mL with respective solvent.	-	*Bacillus subtilis*	1.18	[56]
					*Klebsiella pneumoniae*	1.18	
					*S. aureus*	1.18	
					*Micrococus luteus*	1.18	
					*E. coli*	1.18	
					*Listeria monocytogenes*	2.37	
Maceration	Dried apple without skin (1 g)	80:20 methanol:water (*v*/*v*) (-)	The solution was placed under agitation at 25°C for 1h, followed by filtration. The residue was re-extracted with an additional portion of methanol:water mixture, and the combined extracts were evaporated under reduced pressure.	-	*Acinetobacter baumannii*	>20	[13]
					*E. coli*	5	
					ESBL-*E. coli*	5	
					*K. pneumoniae*	>20	
					ESBL-*K. pneumoniae*	>20	
					*Morganella morganii*	5	
					*P. aeruginosa*	>20	
					*Enterococcus faecalis*	5	
					*L. monocytogenes*	5	
					MRSA	5	
					MSSA	2.5	
Solvent extraction + solid-phase extraction	Apple slices (50 g)	25:75 acetone:ethanol (*v*/*v*) (150 mL)	The solution was blended for 6 min and centrifuged at 0 °C for 25 min. The residue extract was then filtered and concentrated by evaporation under vacuum for 90 min at 40 °C.	-	*S. aureus*	0.05	[57]
					*E. coli*	0.05–0.5	
					*L. monocytogenes*	0.05–50	
					*Salmonella typhimurium*	0.05–0.5	

-not specified in the original study; MIC—minimum inhibitory concentration; ESBL—extended spectrum β-lactamase-producing; MRSA—methicillin-resistant *S. aureus*; MSSA—methicillin-susceptible *S. aureus.*

## Data Availability

Not applicable.

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
