# Peer review of "From Apple Waste to Antimicrobial Solutions: A Review of Phenolics from PGI ‘Maçã de Alcobaça’ and Related Cultivars"

_molecules, 2025, doi:10.3390/molecules30183679_

Round 1

Reviewer 1 Report

Comments and Suggestions for Authors

Line 13, Table 1,

It is not known whether the sample weight is on a dry basis.

Line 193 – Remove  “schematic illustration”, is not necesssary

Line 226  table  3, In the last row of column 5, there is a greater than symbol, is this correct?

Line 402 , Remove a duplicate period. The period only comes after the reference.

Author Response

REVIEW 1

Line 13, Table 1, It is not known whether the sample weight is on a dry basis.

We thank you for your attention. We have adapted the table to include that the results were reported in fresh weight (FW).

Line 193 – Remove “schematic illustration”, is not necessary

We thank you for your suggestion. We have removed “schematic illustration”.

Line 226 Table 3, In the last row of column 5, there is a greater than symbol, is this correct?

Thank you for your observation. The greater than symbol in the last row of column 5 of Table 3 was intended to indicate that quercetin exhibits higher antibacterial activity compared to rutin. To improve clarity for readers, we have revised the text to “quercetin more active than rutin”.

Line 402, Remove a duplicate period. The period only comes after the reference.

We thank you for your attention. We have removed the duplicated period.

Reviewer 2 Report

Comments and Suggestions for Authors

Dear authors, 

The paper of Portugal apple case is suitable for publishing in the journal Molecules

The paper will be interesting to readers in the areas of agriculture, medicine and waste reducing surveys.

Strengths of this review came out the wide description of apple by-products such as apple peel tha represent a valuable source of phenolic compounds.

Through the paper the significant antimicrobial potential, of certain bacteria were described.

In the sample of 'Maçã 27 de Alcobaça', in the case of Portugal, the sustainable strategies for apple waste valorisation was shown.

For practice the concrete evidence is given e.g. its antimicrobial relevance, antibacterial activity, particularly against Gram-positive pathogens such as Staphylococcus aureus that cause great problem in hospitals and medicine generally.  

The part that describes anti-inflammatory, cardioprotective, and metabolic regulatory properties could be more precise, using some more references (e.g insulin sensitivity, reduce oxidative stress, and modulate gut microbiota).

kind regards, the reviewer

Author Response

The paper of Portugal apple case is suitable for publishing in the journal Molecules. The paper will be interesting to readers in the areas of agriculture, medicine and waste reducing surveys. Strengths of this review came out the wide description of apple by-products such as apple peel that represent a valuable source of phenolic compounds. Through the paper the significant antimicrobial potential, of certain bacteria were described. In the sample of 'Maçã 27 de Alcobaça', in the case of Portugal, the sustainable strategies for apple waste valorization were shown. For practice the concrete evidence is given e.g. its antimicrobial relevance, antibacterial activity, particularly against Gram-positive pathogens such as Staphylococcus aureus that cause great problem in hospitals and medicine generally.  We thank you for your positive evaluation of our manuscript and for highlighting the relevance of our work regarding apple by-products, antimicrobial potential, and sustainable strategies.

The part that describes anti-inflammatory, cardioprotective, and metabolic regulatory properties could be more precise, using some more references (e.g insulin sensitivity, reduce oxidative stress, and modulate gut microbiota). We appreciate your valuable comment and suggestion. In the revised version, we have made that section more precise and complemented it with additional references. Specifically, we included studies addressing insulin sensitivity, oxidative stress reduction, and gut microbiota modulation.

Reviewer 3 Report

Comments and Suggestions for Authors

Dear Authors,

The review entitled: “From Apple Waste to Antimicrobial Solutions: A Review of Phenolics from PGI ‘Maçã de Alcobaça’ and Related Cultivars”, could be published in Molecules but after minor revision.

This review focusses on the phenolic profile of 'Maçã de Alcobaça', a Protected Geographical Indication (PGI) apple from Portugal, and its antimicrobial relevance. Their structure–activity relationships and mechanisms of action are critically discussed.

However, the study needs some improvement and additional data to be suitable for publication.

Comments:

The authors could provide MIC values of some positive controls, and comparison between them and main compounds.

A discussion on structure–activity relationships should be more in-depth.

The authors could discuss the toxicity of phlorizin.

Author Response

The review entitled: “From Apple Waste to Antimicrobial Solutions: A Review of Phenolics from PGI ‘Maçã de Alcobaça’ and Related Cultivars”, could be published in Molecules but after minor revision. This review focusses on the phenolic profile of 'Maçã de Alcobaça', a Protected Geographical Indication (PGI) apple from Portugal, and its antimicrobial relevance. Their structure–activity relationships and mechanisms of action are critically discussed. However, the study needs some improvement and additional data to be suitable for publication. We sincerely thank you for your evaluation of our manuscript.

The authors could provide MIC values of some positive controls, and comparison between them and main compounds. We appreciate the reviewer’s comment. We have now added MIC values of positive controls (e.g., ciprofloxacin, vancomycin and gentamicin) and compared them with apple phenolics, highlighting their lower potency but relevant natural activity. This information was included in section 4.1.3.

A discussion on structure–activity relationships should be more in-depth. In the revised version, we have expanded this section by including additional details on the antibacterial activities and structural determinants of chlorogenic acid, catechins, procyanidins, and quercetin.

The authors could discuss the toxicity of phlorizin. We appreciate your suggestion. New information regarding the potential toxicity of phlorizin, including its dose-dependent gastrointestinal effects and the lack of comprehensive clinical safety data, has been added to the manuscript (Section 4.3.).

Reviewer 4 Report

Comments and Suggestions for Authors

The review entitled “From Apple Waste to Antimicrobial Solutions: A Review of Phenolics from PGI ‘Maçã de Alcobaça’ and Related Cultivars” compiles information on apple (Maçã de Alcobaça’variety) waste as a source of phenolic compounds with various bioactivities with application in food, pharmaceutical and other industries. It is generally well-directed and it contributes to expand knowledge on by-products valorization within a circular economy concept, specefically on this Portuguese apple wastes. Thus, I consider it of interest for scientific community and I recommend it for publication after the following minor revisions are addressed:

  • Methodology: Why selected such a short time intervale (Feb-Apr 2025) for conducting the research articles compilation? Despite specifying this dates intervale, most of the cites have been published in previous years (see references section). Check and/or explain.
  • Table 1: Specify in the yield units, with which component is related the ‘kg’ unit: initial raw material submitted to extraction or the obtained extract? (same in table 4). Moreover, specify the ratio sample:solvent employed for the extraction, and other characteristics of the extraction process (temperature, time, stirring…).
  • Line 123: Differences are attributed to extraction methods, so explain how the extraction was conducted in each case, as this review is also focused on describing existing methods for phenolic extraction from these apples.
  • Figure 2: Cite the references from which was collected these data. Could you give some data on proportions for each phenolic in each part of the apples waste?
  • Line 143-144: “In seeds, the most abundant phenolic compounds include chlorogenic acid, (-)-epicatechin, and phloridzin”. This information does not agree with that included in Fig.2.
  • Line 159-161: “Interestingly, certain cultivars such as Fuji, Golden Delicious, Granny Smith, and Pink Lady have been reported to contain higher concentrations of phenolic acids in the flesh compared to the peel”. Why?
  • Line 201-2022: “Phenolics can generate reactive oxygen species (ROS), which lead to oxidative stress and damage cellular structures including proteins, lipids, and DNA.” Under which conditions? Is not this fact opposite to the widely accepted antioxidant potential of the phenolic compounds?
  • Line 209: “Specific phenolics”, specify. Also in line 215.
  • Table 4: Specify parameters selected in each extraction such as temperature, time, ratio solid-to-solvent, power,… Also specify apple variety used for the extraction. Could you provide the name of the identified phenolics in each extraction? It is difficult to obtain conclusions related to the phenolics effect on antibacterial activity if total phenolics is just given for one of the extractions, and no information on individual phenolics is specified.
  • Line 289: the ethyl acetate fraction is not reflected in the table 4.
  • Lines 305-307: Specify which fraction yielded the lowest activity (MIC = 50 mg/mL and give the possible reason why.
  • Line 309: How was obtained this extract? Specify the extraction conditions.
  • Line 318: “In vitro” should be in italic (+ lines 327, 387, 450, 518). Same in line 320 for “in vivo” (+ line 397, 500, 519, 558). Line 408.
  • Line 352: specify which DES were used and their proportions.
  • Lines 359-362: “Taken together, these findings consistently demonstrate that the antioxidant potential of Alcobaça apple cultivars is markedly higher in peel and by-products such as seeds and pomace, aligning with their elevated concentrations of phenolic and flavonoid compounds.” Barely anything is specified in this section about the amount of phenolics in the extracts showing antioxidant activity or even identified phenols. That is essential to stablish clear connections between phenolics and related antioxidant activity.
  • Lines 395-400: This is about antioxidant activity so it should be places in the previous section when talking about that.

Author Response

The review entitled “From Apple Waste to Antimicrobial Solutions: A Review of Phenolics from PGI ‘Maçã de Alcobaça’ and Related Cultivars” compiles information on apple (Maçã de Alcobaça’variety) waste as a source of phenolic compounds with various bioactivities with application in food, pharmaceutical and other industries. It is generally well-directed, and it contributes to expand knowledge on by-products valorization within a circular economy concept, specefically on this Portuguese apple wastes. Thus, I consider it of interest for scientific community, and I recommend it for publication after the following minor revisions are addressed:

  • Methodology: Why selected such a short time intervale (Feb-Apr 2025) for conducting the research articles compilation? Despite specifying this dates intervale, most of the cites have been published in previous years (see references section). Check and/or explain. The time interval indicated (February–April 2025) refers to the period during which we carried out the compilation and writing of the review, and not to the publication dates of the studies included. As expected, most of the cited works were published in previous years, since our aim was to provide a comprehensive overview of the available literature on this topic.
  • Table 1: Specify in the yield units, with which component is related the ‘kg’ unit: initial raw material submitted to extraction or the obtained extract? (same in table 4). Moreover, specify the ratio sample:solvent employed for the extraction, and other characteristics of the extraction process (temperature, time, stirring…). We thank you for this valuable comment. In Table 1 and Table 4, we have now specified that the yield values are expressed as g GAE kg⁻¹ FW, indicating that the unit “kg” refers to the fresh weight of the initial raw material submitted to extraction. In addition, we included in the table footnotes the sample:solvent ratio employed, as well as the main extraction conditions (temperature, time, and stirring), to provide a clearer understanding of the process.
  • Line 123: Differences are attributed to extraction methods, so explain how the extraction was conducted in each case, as this review is also focused on describing existing methods for phenolic extraction from these apples. We have now specified the extraction methods used in each study, which clarifies that differences in phenolic yield may be due to both cultivar/environmental factors and extraction methods.
  • Figure 2: Cite the references from which was collected these data. Could you give some data on proportions for each phenolic in each part of the apples waste? We have now added the relevant references for the data presented, as well as the reported values of total phenolic content for each part of the apples. Detailed proportions of individual phenolic compounds in each part were not included, as these were not consistently reported in the original studies.
  • Line 143-144: “In seeds, the most abundant phenolic compounds include chlorogenic acid, (-)-epicatechin, and phloridzin”. This information does not agree with that included in Fig.2. Thank you for pointing this out. The figure has been updated to reflect the correct information regarding the most abundant phenolic compounds in seeds, ensuring consistency with the text.
  • Line 159-161: “Interestingly, certain cultivars such as Fuji, Golden Delicious, Granny Smith, and Pink Lady have been reported to contain higher concentrations of phenolic acids in the flesh compared to the peel”. Why? The observed higher concentrations of phenolic acids in the flesh of certain cultivars may be related to cultivar-specific metabolic distribution of phenolic compounds, differences in tissue composition, or physiological roles of these compounds in the fruit. We have added a brief explanation in the text to address this point.
  • Line 201-2022: “Phenolics can generate reactive oxygen species (ROS), which lead to oxidative stress and damage cellular structures including proteins, lipids, and DNA.” Under which conditions? Is not this fact opposite to the widely accepted antioxidant potential of the phenolic compounds? We thank you for your attention. The text has been revised to clarify that the pro-oxidant activity of phenolics occurs only under specific conditions, making this effect context-dependent rather than contradictory to their antioxidant role.
  • Line 209: “Specific phenolics”, specify. Also, in line 215. We have now specified the phenolic compounds reported to downregulate antibiotic resistance genes or other resistance determinants and exert bifidogenic effects.
  • Table 4: Specify parameters selected in each extraction such as temperature, time, ratio solid-to-solvent, power, … Also specify apple variety used for the extraction. Could you provide the name of the identified phenolics in each extraction? It is difficult to obtain conclusions related to the phenolics effect on antibacterial activity if total phenolics is just given for one of the extractions, and no information on individual phenolics is specified. We thank the reviewer for the comment. Table 4 is intended to compare extraction methods in terms of their antibacterial activity rather than to provide full phenolic profiles. Total phenolic yield is reported only where available in the original studies, and detailed information on individual phenolics is not consistently provided. The focus is on how extraction parameters affect antibacterial effects.
  • Line 289: the ethyl acetate fraction is not reflected in the table 4. The ethyl acetate fraction is already included in Table 4, as indicated in the first row.
  • Lines 305-307: Specify which fraction yielded the lowest activity (MIC = 50 mg/mL and give the possible reason why. The fraction that yielded the lowest antibacterial activity (MIC = 50 mg/mL) against Listeria monocytogenes was fraction IV. This lower activity may be related to its specific phenolic composition, as some phenolic compounds present in other fractions that contribute more strongly to antibacterial effects might be less abundant or absent in fraction IV. The manuscript was altered accordingly.
  • Line 309: How was obtained this extract? Specify the extraction conditions. The study by Friedman et al. (2013) evaluated the bactericidal activities of apple skin extracts against various foodborne pathogens. However, the article does not provide detailed information on the extraction methods or conditions used to obtain these apple skin extracts.
  • Line 318: “In vitro” should be in italic (+ lines 327, 387, 450, 518). Same in line 320 for “in vivo” (+ line 397, 500, 519, 558). Line 408. We have carefully revised the manuscript and corrected the formatting, placing in vitro and in vivo in italics in all the indicated lines.
  • Line 352: specify which DES were used and their proportions. We have now specified the deep eutectic solvent (DES) used by Bottu et al., indicating that it was composed of choline chloride and ethylene glycol in a 1:4 molar ratio, and highlighted its effectiveness for extracting bioactive compounds from apple pomace.
  • Lines 359-362: “Taken together, these findings consistently demonstrate that the antioxidant potential of Alcobaça apple cultivars is markedly higher in peel and by-products such as seeds and pomace, aligning with their elevated concentrations of phenolic and flavonoid compounds.” Barely anything is specified in this section about the amount of phenolics in the extracts showing antioxidant activity or even identified phenols. That is essential to stablish clear connections between phenolics and related antioxidant activity. We thank the reviewer for the comment. We have revised the section to include more specific information on the phenolic content of the extracts showing antioxidant activity, as reported in the original studies. These additions clarify the relationship between phenolic composition and the observed antioxidant effects.
  • Lines 395-400: This is about antioxidant activity so it should be places in the previous section when talking about that. We agree with your suggestion, and the content referring to antioxidant activity (lines 395–400) has been moved to the previous section where antioxidant properties are discussed.